# Debiasing Concept-based Explanations with Causal Analysis

**Mohammad Taha Bahadori,  David E. Heckerman**

`{bahadorm, heckerma}@amazon.com`

## Abstract

Concept-based explanation approach is a popular model interpertability tool because it expresses the reasons for a model's predictions in terms of concepts that are meaningful for the domain experts. In this work, we study the problem of the concepts being correlated with confounding information in the features. We propose a new causal prior graph for modeling the impacts of unobserved variables and a method to remove the impact of confounding information and noise using a two-stage regression technique borrowed from the instrumental variable literature. We also model the completeness of the concepts set and show that our debiasing method works when the concepts are not complete. Our synthetic and real-world experiments demonstrate the success of our method in removing biases and improving the ranking of the concepts in terms of their contribution to the explanation of the predictions.

## 1 Introduction

Explaining the predictions of neural networks through higher level concepts (Kim et al., 2018; Ghorbani et al., 2019; Brocki & Chung, 2019; Hamidi-Haines et al., 2018) enables model interpretation on data with complex manifold structure such as images. It also allows the use of domain knowledge during the explanation process. The concept-based explanation has been used for medical imaging (Cai et al., 2019), breast cancer histopathology (Graziani et al., 2018), cardiac MRIs (Clough et al., 2019), and meteorology (Sprague et al., 2019).

When the set of concepts is carefully selected, we can estimate a model in which the discriminative information flow from the feature vectors $\mathbf{x}$ through the concept vectors $\mathbf{c}$ and reach the labels $\mathbf{y}$. To this end, we train two models for prediction of the concept vectors from the features denoted by $\widehat{\mathbf{c}}(\mathbf{x})$ and the labels from the predicted concept vector $\widehat{\mathbf{y}}(\widehat{\mathbf{c}})$. This estimation process ensures that for each prediction we have the reasons for the prediction stated in terms of the predicted concept vector $\widehat{\mathbf{c}}(\mathbf{x})$.

However, in reality, noise and confounding information (due to e.g. non-discriminative context) can influence both of the feature and concept vectors, resulting in confounded correlations between them. Figure 1 provides an evidence for noise and confounding in the CUB-200-2011 dataset (Wah et al., 2011). We train two predictors for the concepts vectors based on features $\widehat{\mathbf{c}}(\mathbf{x})$ and labels $\widehat{\mathbf{c}}(\mathbf{y})$ and compare the Spearman correlation coefficients between their predictions and the true ordinal value of the concepts. Having concepts for which $\widehat{\mathbf{c}}(\mathbf{x})$ is more accurate than $\widehat{\mathbf{c}}(\mathbf{y})$ could be due to noise, or due to hidden variables independent of the labels that spuriously correlated $\mathbf{c}$ and $\mathbf{x}$, leading to undesirable explanations that include confounding or noise.

In this work, using the Concept Bottleneck Models (CBM) (Koh et al., 2020; Losch et al., 2019) we demonstrate a method for removing the counfounding and noise (debiasing) the explanation with concept vectors and extend the results to Testing with Concept Activation Vectors (TCAV) (Kim et al., 2018) technique. We provide a new causal prior graph to account for the confounding information and concept completeness (Yeh et al., 2020). We describe the challenges in estimation of our causal prior graph and propose a new learning procedure. Our estimation technique defines and predicts debiased concepts such that the predictive information of the features maximally flow through them.

We show that using a two-stage regression technique from the instrumental variables literature, we can successfully remove the impact of the confounding and noise from the predicted concept vectors. Our proposed procedure has three steps: (1) debias the concept vectors using the labels, (2) predict

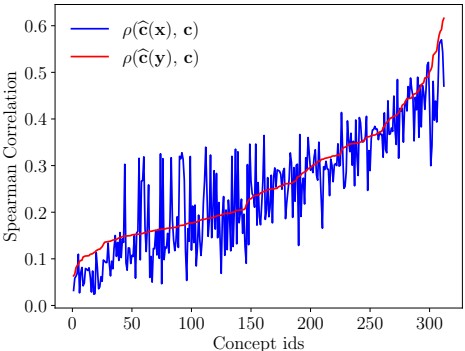

Figure 1: Spearman correlation coefficients ($\rho$) of the predictors of the concepts given features $\widehat{\mathbf{c}}(\mathbf{x})$ and labels $\widehat{\mathbf{c}}(\mathbf{y})$ for the 312 concepts in the test partition of the CUB-200-2011 dataset (Wah et al., 2011). 112 concepts can be predicted more accurately with the features rather than the labels. Concept ids in the x-axis are sorted in the increasing $\rho(\widehat{\mathbf{c}}(\mathbf{y}), \mathbf{c})$ order. We provide the detailed steps to obtain the figure in Section 4.2.

the debiased concept vectors using the features, and (3) use the predict concept vectors in the second step to predict the labels. Optionally, we can also find the residual predictive information in the features that are not in the concepts.

We validate the proposed method using a synthetic dataset and the CUB-200-2011 dataset. On the synthetic data, we have access to the ground truth and show that in the presence of confounding and noise, our debiasing procedure improves the accuracy of recovering the true concepts. On the CUB-200-2011 dataset, we use the RemOve And Retrain (ROAR) framework (Hooker et al., 2019) to show that our debiasing procedure ranks the concepts in the order of their explanation more accurately than the regular concept bottleneck models. We also show that we improve the accuracy of CBNs in the prediction of labels using our debiasing technique. Finally, using several examples, we also qualitatively show when the debasing helps improve the quality of concept-based explanations.

## 2 METHODOLOGY

**Notations.** We follow the notation of Goodfellow et al. (2016) and denote random vectors by bold font letters $\mathbf{x}$ and their values by bold symbols $\boldsymbol{x}$. The notation $p(\mathbf{x})$ is a probability measure on $\mathbf{x}$ and $\mathrm{d}p(\mathbf{x} = \boldsymbol{x})$ is the infinitesimal probability mass at $\mathbf{x} = \boldsymbol{x}$. We use $\widehat{\mathbf{y}}(\mathbf{x})$ to denote the the prediction of $\mathbf{y}$ given $\mathbf{x}$. In the graphical models, we show the observed and unobserved variables using filled and hollow circles, respectively.

**Problem Statement.** We assume that during the training phase, we are given triplets $(\boldsymbol{x}_i, \boldsymbol{c}_i, \boldsymbol{y}_i)$ for $i = 1, \ldots, n$ data points. In addition to the regular features $\boldsymbol{x}$ and labels $\boldsymbol{y}$, we are given a human interpretable concepts vector $\boldsymbol{c}$ for each data point. Each element of the concept vector measures the degree of existence of the corresponding concept in the features. Thus, the concept vector typically have binary or ordinal values. Our goal is to learn to predict $\boldsymbol{y}$ as a function of $\boldsymbol{x}$ and use $\boldsymbol{c}$ for explaining the predictions. Performing in two steps, we first learn a function $\widehat{\mathbf{c}}(\mathbf{x})$ and then learn another function $\widehat{\mathbf{y}}(\widehat{\mathbf{c}}(\mathbf{x}))$. The prediction $\widehat{\mathbf{c}}(\mathbf{x})$ is the explanation for our prediction $\widehat{\mathbf{y}}$. During the test time, only the features are given and the prediction+explanation algorithm predicts both $\widehat{\mathbf{y}}$ and $\widehat{\mathbf{c}}$.

In this paper, we aim to remove the bias and noise components from the estimated concept vector $\widehat{\mathbf{c}}$ such that it explains the reasons for prediction of the labels more accurately. To this end, we first need to propose a new causal prior graph that includes the potential unobserved confounders.

### 2.1 A NEW CAUSAL PRIOR GRAPH FOR CBMS

Figure 2a shows the ideal situation in explanation via high-level concepts. The generative model corresponding to Figure 2a states that for generating each feature $\boldsymbol{x}_i$ we first randomly draw the label $\boldsymbol{y}_i$. Given the label, we draw the concepts $\boldsymbol{c}_i$. Given the concepts, we generate the features. The

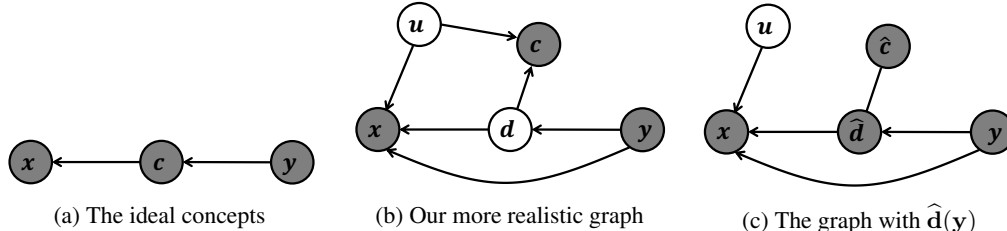

(a) The ideal concepts    (b) Our more realistic graph    (c) The graph with $\widehat{\mathbf{d}}(\mathbf{y})$

Figure 2: (a) The ideal view of the causal relationships between the features $\mathbf{x}$, concepts $\mathbf{c}$, and labels $\mathbf{y}$. (b) In a more realistic setting, the unobserved confounding variable $\mathbf{u}$ impacts both $\mathbf{x}$ and $\mathbf{c}$. The shared information between $\mathbf{x}$ and $\mathbf{y}$ go through the discriminative part of the concepts $\mathbf{d}$. We also model the completeness of the concepts via a direct edge from the features $\mathbf{x}$ to the labels $\mathbf{y}$. (c) When we use $\widehat{\mathbf{d}}(\mathbf{y}) = E[\mathbf{c}|\mathbf{y}]$ in place of $\mathbf{d}$ and $\mathbf{c}$, we eliminate the confounding link $\mathbf{u} \to \mathbf{c}$.

hierarchy in this graph is from nodes with less detailed information (labels) to more detailed ones (features, images).

This model in Figure 2a is an explanation for the phenomenon in Figure 1, because the noise in generation of the concepts allows the $\mathbf{x}$—$\mathbf{c}$ edge to be stronger than the $\mathbf{c}$—$\mathbf{y}$ edge. However, another (non-mutually exclusive) explanation for this phenomenon is the existence of hidden confounders $\mathbf{u}$ shown in Figure 2b. In this graphical model, $\mathbf{u}$ represents the confounders and $\mathbf{d}$ represents the unconfounded concepts. Note that we assume that the confounders $\mathbf{u}$ and labels $\mathbf{y}$ are independent when $\mathbf{x}$ and $\mathbf{c}$ are not observed.

Another phenomenon captured in Figure 2b is the lack of concept completeness (Yeh et al., 2020). It describes the situation when the features, compared to the concepts, have additional predictive information about the labels.

The non-linear structural equations corresponding to the causal prior graph in Figure 2b are as follows

$$\mathbf{d} = \boldsymbol{f}_1(\mathbf{y}) + \boldsymbol{\varepsilon}_d, \tag{1}$$

$$\mathbf{c} = \mathbf{d} + \boldsymbol{h}(\mathbf{u}), \tag{2}$$

$$\mathbf{x} = \boldsymbol{f}_2(\mathbf{u}, \mathbf{d}) + \boldsymbol{f}_3(\mathbf{y}) + \boldsymbol{\varepsilon}_x, \tag{3}$$

for some vector functions $\boldsymbol{h}, \boldsymbol{f}_1, \boldsymbol{f}_2$, and $\boldsymbol{f}_3$. We have $\boldsymbol{\varepsilon}_d \perp\!\!\!\perp \mathbf{y}$ and $\mathbf{u} \perp\!\!\!\perp \mathbf{y}$. Our definition of $\mathbf{d}$ in Eq. (2) does not restrict $\mathbf{u}$, because we simply attribute the difference between $\mathbf{c}$ and $\boldsymbol{f}_1(\mathbf{y})$ to a function of the latent confounder $\mathbf{u}$ and noise.

Our causal prior graph in Figure 2b corresponds to a generative process in which to generate an observed triplet $(\boldsymbol{x}_i, \boldsymbol{c}_i, \boldsymbol{y}_i)$ we first draw a label $\boldsymbol{y}_i$ and a confounder $\boldsymbol{u}_i$ vector independently. Then we draw the discriminative concepts $\boldsymbol{d}_i$ based on the label and generate the features $\boldsymbol{x}_i$ jointly based on the concepts, label, and the confounder. Finally, we draw the observed concept vector $\boldsymbol{c}_i$ based on the drawn concept and confounder vectors.

Both causal graphs reflect our assumption that the direction of causality is from the labels to concepts and then to the features, $\mathbf{y} \to \mathbf{d} \to \mathbf{x}$, to ensure that $\mathbf{u}$ and $\mathbf{y}$ are marginally independent in Figure 2b. This direction also correspond to moving from more abstract class labels to concepts to detailed features. During estimation, we fit the functions in the $\mathbf{x} \to \mathbf{d} \to \mathbf{y}$ direction, because finding the statistical strength of an edge does not depend on its direction.

Estimation of the model in Figure 2b is challenging because there are two distinct paths for the information from the labels $\mathbf{y}$ to reach the features $\mathbf{x}$. Our solution is to prioritize the bottleneck path and estimate the $\mathbf{y} \to \mathbf{d} \to \mathbf{x}$, then estimate the residuals of the regression using the $\mathbf{y} \to \mathbf{x}$ direct path. Our two-stage estimation technique ensures that the predictive information of the features maximally flow through the concepts. In the next sections, we focus on the first phase and using a two-stage regression technique borrowed from the instrumental variables literature to eliminate the noise and confounding in estimation of the $\mathbf{d} \to \mathbf{x}$ link.

---

**Algorithm 1** Debiased CBMs

---

**Require:** Data tuples $(\boldsymbol{x}_i, \boldsymbol{c}_i, \boldsymbol{y}_i)$ for $i = 1, \ldots, n$.
1: Estimate a model $\widehat{\mathbf{d}}(\mathbf{y}) = E[\mathbf{c}|\mathbf{y}]$ using $(\boldsymbol{c}_i, \boldsymbol{y}_i)$ pairs.
2: Train a neural network as an estimator for $p_{\widehat{\phi}}(\mathbf{d}|\mathbf{x})$ using $(\boldsymbol{x}_i, \widehat{\boldsymbol{d}}_i))$ pairs.
3: Use pairs $(\boldsymbol{x}_i, \boldsymbol{y}_i)$ to estimate function $\boldsymbol{g}_{\boldsymbol{\theta}}$ by fitting $\int \boldsymbol{g}_{\boldsymbol{\theta}}(\boldsymbol{d}) \mathrm{d} p_{\widehat{\phi}}(\mathbf{d} = \boldsymbol{d}|\boldsymbol{x}_i)$ to $\boldsymbol{y}_i$.
4: Compute the debiased explanations $E[\mathbf{d}|\boldsymbol{x}_i] - \frac{1}{n} \sum_{i=1}^{n} E[\mathbf{d}|\boldsymbol{x}_i]$ for $i = 1, \ldots, n$.
5: **return** The CBM defined by $(p_{\widehat{\phi}}, \boldsymbol{g}_{\boldsymbol{\theta}})$ and the debiased explanations.

---

## 2.2 INSTRUMENTAL VARIABLES

**Background on two-stage regression.** In causal inference, instrumental variables (Stock, 2015; Pearl, 2009) denoted by $\mathbf{z}$ are commonly used to find the causal impact of a variable $\mathbf{x}$ on $\mathbf{y}$ when $\mathbf{x}$ and $\mathbf{y}$ are jointly influenced by an unobserved confounder $\mathbf{u}$ (i.e., $\mathbf{x} \leftarrow \mathbf{u} \rightarrow \mathbf{y}$). The key requirement is that $\mathbf{z}$ should be correlated with $\mathbf{x}$ but independent of the confounding variable $\mathbf{u}$ (i.e. $\mathbf{z} \rightarrow \mathbf{x} \rightarrow \mathbf{y}$ and $\mathbf{z} \perp\!\!\!\perp \mathbf{u}$). The commonly used 2-stage least squares first regresses $\mathbf{x}$ in terms of $\mathbf{z}$ to obtain $\widehat{\mathbf{x}}$ followed by regression of $\mathbf{y}$ in terms of $\widehat{\mathbf{x}}$. Because of independence between $\mathbf{z}$ and $\mathbf{u}$, $\widehat{\mathbf{x}}$ is also independent of $\mathbf{u}$. Thus, in the second regression the confounding impact of $\mathbf{u}$ is eliminated. Our goal is to use the two-stage regression trick again to remove the confounding factors impacting features and concept vectors. The instrumental variable technique can be used for eliminating the biases due to the measurement errors (Carroll et al., 2006).

**Two-Stage Regression for CBMs.** In our causal graph in Figure 2b, the label $\mathbf{y}$ can be used for the study of the relationship between concepts $\mathbf{d}$ and features $\mathbf{x}$. We predict $\mathbf{d}$ as a function of $\mathbf{y}$ and use it in place of the concepts in the concept bottleneck models. The graphical model corresponding to this procedure is shown in Figure 2c, where the link $\mathbf{u} \rightarrow \mathbf{c}$ is eliminated. In particular, given the independence relationship $\mathbf{y} \perp\!\!\!\perp \mathbf{u}$, we have $\widehat{\mathbf{d}}(\mathbf{y}) = E[\mathbf{c}|\mathbf{y}] \perp\!\!\!\perp \boldsymbol{h}(\mathbf{u})$. This is the basis for our debiasing method in the next section.

## 2.3 THE ESTIMATION METHOD

Our estimation uses the observation that in graph 2b the label vector $\mathbf{y}$ is a valid instrument for removing the correlations due to $\mathbf{u}$. Combining Eqs. (1) and (2) we have $\mathbf{c} = \boldsymbol{f}_1(\mathbf{y}) + \boldsymbol{h}(\mathbf{u}) + \varepsilon_d$. Taking expectation with respect to $p(\mathbf{c}|\mathbf{y})$, we have

$$E[\mathbf{c}|\mathbf{y}] = E[\boldsymbol{f}_1(\mathbf{y}) + \boldsymbol{h}(\mathbf{u}) + \varepsilon_d|\mathbf{y}] = \boldsymbol{f}_1(\mathbf{y}) + E[\boldsymbol{h}(\mathbf{u})] + E[\varepsilon_d]. \tag{4}$$

The last step is because both $\mathbf{u}$ and $\varepsilon_d$ are independent of $\mathbf{y}$. Thus, two terms are constant in terms of $\mathbf{x}$ and $\mathbf{y}$ and can be eliminated after estimation. Eq. (4) allows us to remove the impact of $\mathbf{u}$ and $\varepsilon_d$ and estimate the denoised and debiased $\widehat{\mathbf{d}}(\mathbf{y}) = E[\mathbf{c}|\mathbf{y}]$. We find $E[\mathbf{c}|\mathbf{y}]$ using a neural network trained on $(\boldsymbol{c}_i, \boldsymbol{y}_i)$ pairs and use them as pseudo-observations in place of $\boldsymbol{d}_i$. Given our debiased prediction for the discriminative concepts $\boldsymbol{d}_i$, we can perform the CBMs' two-steps of $\mathbf{x} \rightarrow \mathbf{d}$ and $\mathbf{d} \rightarrow \mathbf{y}$ estimation.

Because we use expected values of $\mathbf{c}$ in place of $\mathbf{d}$ during the learning process (i.e., $\widehat{\mathbf{d}}(\mathbf{y}) = E[\mathbf{c}|\mathbf{y}]$), the debiased concept vectors have values within the ranges of original concept vectors $\mathbf{c}$. Thus, we do not lose the human readability with the debiased concept vectors.

**Incorporating Uncertainty in Prediction of Concepts.** Our empirical observations show that prediction of the concepts from the features can be highly uncertain. Hence, we present a CBM estimator that takes into account the uncertainties in prediction of the concepts. We take the conditional expectation of the labels $\mathbf{y}$ given features $\mathbf{x}$ as follows

$$E[\mathbf{y}|\mathbf{x}] = E[\boldsymbol{g}_{\boldsymbol{\theta}}(\widehat{\mathbf{d}})|\mathbf{x}] = \int \boldsymbol{g}_{\boldsymbol{\theta}}(\boldsymbol{d}) \mathrm{d} p_{\phi}(\mathbf{d} = \boldsymbol{d}|\mathbf{x}), \tag{5}$$

where $p_{\phi}(\mathbf{d}|\mathbf{x})$ is the probability function, parameterized by $\phi$, that captures the uncertainty in prediction of labels from features. The $\boldsymbol{g}_{\boldsymbol{\theta}}(\cdot)$ function predicts labels from the debiased concepts.

In summary, we perform the steps in Algorithm 1 to learn debiased CBMs. In Line 3, we approximate the integral using Monte Carlo approach by drawing from the distribution $p_{\widehat{\phi}}(\mathbf{d}|\mathbf{x})$ estimated in Line 2, see (Hartford et al., 2017). Given the labels' data type, we can use the appropriate loss functions and are not limited to the least squares loss. Line 4 removes the constant mean that can be due to noise or confounding, as we discussed after Eq. (4). In the common case of binary concept vectors, we use the debiased concepts estimated in Line 4 to rank the concepts in the order of their contribution to explanation of the predictions.

## 3  DISCUSSIONS

**Debiasing TCAVs.**    While we presented our debiasing algorithm for the CBMs, we can easily use it to debias the TCAV (Kim et al., 2018) explanations too. We can use the first step to remove the bias due to the confounding and perform TCAV using $\widehat{\boldsymbol{d}}$ vectors, instead of $\boldsymbol{c}$ vectors. The TCAV method is attractive, because unlike CBMs, it analyzes the existing neural networks and does not need to define a new model.

**Measuring Concept Completeness.**    Our primary objective of modeling the concept incompleteness phenomenon in our causal prior graph is to show that our debiasing method does not need the concept completeness assumption to work properly. If we are interested in measuring the degree of completeness of the concepts, we can do it based on the definition by Yeh et al. (2020). To this end, we fit a unrestricted neural network $\boldsymbol{q}(\mathbf{x}) = E[\mathbf{y}|\mathbf{x}]$ to the to the residuals in the step 3 of our debiasing algorithm. The function $\boldsymbol{q}(\cdot)$ captures the residual information in $\mathbf{x}$. We compare the improvement in prediction accuracy over the accuracy in step 3 to quantify the degree of concept incompleteness. Because we first predict the labels $\mathbf{y}$ using the CBM and then fit the $\boldsymbol{q}(\mathbf{x})$ to the residuals, we ensure that the predictive information maximally go through the CBM link $\mathbf{x} \leftarrow \mathbf{d} \leftarrow \mathbf{y}$.

**Prior Work on Causal Concept-Based Explanation.**    Among the existing works on causal concept-based explanation, Goyal et al. (2019) propose a different causal prior graph to model the spurious correlations among the concepts and remove them using conditional variational auto-encoders. In contrast, we aim at handling noise and spurious correlations between the features and concepts using the labels as instruments. Which work is more appropriate for a problem depending on the assumptions underlying that problem.

**The Special Linear Gaussian Case.**    When the concepts have real continuous values, we might use a simple multivariate Gaussian distribution to model $p(\mathbf{d}|\mathbf{x}) = \mathcal{N}(\mathbf{x}, \sigma\boldsymbol{I})$, $\sigma > 0$. If we further use a linear regression to predict labels from the concepts, we can show that the steps are simplified as follows:

1. Learn $\widehat{\mathbf{d}}(\mathbf{y})$ by predicting $\boldsymbol{y}_i \rightarrow \boldsymbol{c}_i$.

2. Learn $\widehat{\widehat{\mathbf{d}}}(\mathbf{x})$ by predicting $\boldsymbol{x}_i \rightarrow \widehat{\boldsymbol{d}}_i$.

3. Learn $\widehat{\mathbf{y}}(\widehat{\widehat{\mathbf{d}}})$ by predicting $\widehat{\widehat{\boldsymbol{d}}}_i \rightarrow \boldsymbol{y}_i$.

The above steps show that given the linear Gaussian assumptions, steps 2 and 3 coincide with the *sequential bottleneck* (Koh et al., 2020) training of CBMs. We only need to change the concepts from $\boldsymbol{c}_i$ to $\widehat{\boldsymbol{d}}_i$ we can eliminate the noise and confounding bias.

## 4  EXPERIMENTS

Evaluation of explanation algorithms is notoriously difficult. Thus, we first present experiments with synthetic data to show that our debiasing technique improves the explanation accuracy when we know the true explanations. Then, on the CUB-200-2011 dataset, we use the ROAR (Hooker et al., 2019) framework to show that the debiasing improves the explanation accuracy. Finally, using several examples, we identify the circumstances in which the debiasing helps.

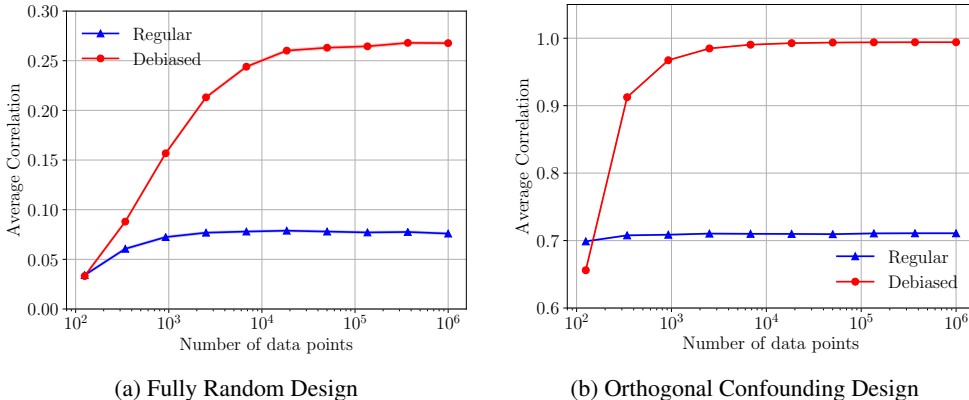

(a) Fully Random Design      (b) Orthogonal Confounding Design

Figure 3: Correlation between the estimated concept vectors and the true discriminative concept vectors as the number of data points grow. Notice the different ranges of the y-axes and the logarithmic scale of the x-axes.

## 4.1 SYNTHETIC DATA EXPERIMENTS

We create a synthetic dataset according to the following steps:

1. Generate $n$ vectors $\boldsymbol{y}_i \in \mathbb{R}^{100}$ with elements distributed according to unit normal distribution $\mathcal{N}(0,1)$.

2. Generate $n$ vectors $\boldsymbol{u}_i \in \mathbb{R}^{100}$ with elements distributed according to unit normal distribution $\mathcal{N}(0,1)$.

3. Generate $n$ vectors $\boldsymbol{\varepsilon}_{c,i} \in \mathbb{R}^{100}$ with elements distributed according to scaled normal distribution $\mathcal{N}(0, \sigma = 0.02)$.

4. Generate $n$ vectors $\boldsymbol{\varepsilon}_{x,i} \in \mathbb{R}^{100}$ with elements distributed according to scaled normal distribution $\mathcal{N}(0, \sigma = 0.02)$.

5. Generate matrices $\boldsymbol{W}_1, \boldsymbol{W}_2, \boldsymbol{W}_3, \boldsymbol{W}_4 \in \mathbb{R}^{100 \times 100}$ with elements distributed according to scaled normal distribution $\mathcal{N}(0, \sigma = 0.1)$.

6. Compute $\boldsymbol{d}_i = \boldsymbol{W}_1 \boldsymbol{y}_i + \boldsymbol{\varepsilon}_{d,i}$ for $i = 1, \ldots, n$.

7. Compute $\boldsymbol{c}_i = \boldsymbol{d}_i + \boldsymbol{W}_2 \boldsymbol{u}_i$ for $i = 1, \ldots, n$.

8. Compute $\boldsymbol{x}_i = \boldsymbol{W}_3 \boldsymbol{d}_i + \boldsymbol{W}_4 \boldsymbol{u}_i + \boldsymbol{\varepsilon}_{x,i}$ for $i = 1, \ldots, n$.

In Figure 3a, we plot the correlation between the true unconfounded and noiseless concepts $\boldsymbol{W}\mathbf{y}$ and the estimated concept vectors with the regular two-step procedure (without debiasing) and our debiasing method, as a function of sample size $n$. The results show that the bias due to confounding does not vanish as we increase the sample size and our debiasing technique can make the results closer to the true discriminative concepts.

In the ideal case, when the confounding impact is orthogonal to the impact of the labels, our debiasing algorithm recovers the true concepts more accurately. Figure 3b demonstrates the scenario when $\boldsymbol{W}_1 \perp \boldsymbol{W}_2, \boldsymbol{W}_4$ in the synthetic data generation. To generate matrices with this constraint, we first generate a unitary matrix $\boldsymbol{Q}$ and generate four diagonal matrices $\boldsymbol{\Lambda}_1, \ldots, \boldsymbol{\Lambda}_4$ with diagonal elements drawn from $\mathrm{Uniform}(0.1, 1)$. This choice of the distribution for the diagonal elements caps the condition number of the matrices by 10. To satisfy the orthogonality constraint, we set the first 50 diagonal elements of $\boldsymbol{\Lambda}_2$ and $\boldsymbol{\Lambda}_4$ and last 50 diagonal elements of $\boldsymbol{\Lambda}_1$ to zero. We compute the matrices as $\boldsymbol{W}_j = \boldsymbol{Q}\boldsymbol{\Lambda}_j\boldsymbol{Q}^\top$ for $j = 1, \ldots, 4$. The orthogonality allows perfect separation of $\mathbf{u}$ and $\mathbf{y}$ impacts and the perfect debiasing by our method as the sample size grows.

In Figure 6, we repeat the synthetic experiments in a higher noise regime where the standard deviation of the noise variables $\boldsymbol{\varepsilon}_c$ and $\boldsymbol{\varepsilon}_c$ is set to $\sigma = 0.1$. The results confirm our main claim about our debiasing algorithm.

Table 1: Mapping the concept annotations to real values.

| Annotation | Certainty | Ordinal Score | Numeric Map |
|---|---|---|---|
| Doesn't Exist | definitely | 0 | 0 |
| Doesn't Exist | probably | 1 | 1/6 |
| Doesn't Exist | guessing | 2 | 2/6 |
| Doesn't Exist | not visible | 3 | 3/6 |
| Exists | not visible | 3 | 3/6 |
| Exists | guessing | 4 | 4/6 |
| Exists | probably | 5 | 5/6 |
| Exists | definitely | 6 | 1 |

## 4.2 CUB-200-2011 EXPERIMENTS

**Dataset and preprocessing.** We evaluate the performance of the proposed approach on the CUB-200-2011 dataset (Wah et al., 2011). The dataset includes 11788 pictures (in 5994/5794 train/test partitions) of 200 different types of birds, annotated both for the bird type and 312 different concepts about each picture. The concept annotations are binary, whether the concept exists or not. However, for each statement, a four-level certainty score has been also assigned: 1: not visible, 2: guessing, 3: probably, and 4: definitely. We combine the binary annotation and the certainty score to create a 7-level ordinal variable as the annotation for each image as summarized in Table 1. For simplicity, we map the 7-level ordinal values to uniformly spaced valued in the $[0, 1]$ interval. We randomly choose 15% of the training set and hold out as the validation set.

**The result in Figure 1.** To compare the association strength between $\mathbf{y}$ and $\mathbf{c}$ with the association strength between $\mathbf{x}$ and $\mathbf{c}$ we train two predictors of concepts $\widehat{\mathbf{c}}(\mathbf{x})$ and $\widehat{\mathbf{c}}(\mathbf{y})$. We use TorchVision's pre-trained ResNet152 network (He et al., 2016) for prediction of the concepts from the images. Because the labels $\mathbf{y}$ have categorical values, $\widehat{\mathbf{c}}(\mathbf{y})$ is simply the average concept annotation scores per each class. We use the Spearman correlation to find the association strengths in pairs $(\widehat{\mathbf{c}}(\mathbf{x}), \mathbf{c})$ and $(\widehat{\mathbf{c}}(\mathbf{y}), \mathbf{c})$ because the concept annotations are ordinal numbers. The concept ids in the x-axis are sorted in terms of increasing values of $\rho(\widehat{\mathbf{c}}(\mathbf{y}), \mathbf{c})$.

The top ten concepts with the largest values of $\rho(\widehat{\mathbf{c}}(\mathbf{x}), \mathbf{c}) - \rho(\widehat{\mathbf{c}}(\mathbf{y}), \mathbf{c})$ are 'has back color::green', 'has upper tail color::green', 'has upper tail color::orange', 'has upper tail color::pink', 'has back color::rufous', 'has upper tail color::purple', 'has back color::pink', 'has upper tail color::iridescent', 'has back color::purple', and 'has back color::iridescent'. These concepts are all related to color and can be easily confounded by the context of the images.

**Training details for Algorithm 1.** We model the distribution of the concept logits as Gaussians with means equal to the ResNet152's logit outputs and a diagonal covariance matrix. We estimate the variance for each concept by using the logits of the true concept annotation scores that are clamped into $[0.05, 0.95]$ to avoid large logit numbers. In each iteration of the training loop for Line 3, we draw 25 samples from the estimated $p(\mathbf{d}|\mathbf{x})$. Predictor of labels from concepts (the function $\boldsymbol{g}(\cdot)$ in Eq. (5)) is a three-layer feed-forward neural network with hidden layer sizes (312, 312, 200). There is a skip connection from the input to the penultimate layer. All algorithms are trained with Adam optimization algorithm (Kingma & Ba, 2014).

**Quantitative Results.** Comparing to the baseline algorithm, our debiasing technique increases the average Spearman correlation between $\widehat{\mathbf{c}}(\mathbf{x})$ and $\widehat{\mathbf{c}}(\mathbf{y})$ from 0.406 to 0.508. For the above 10 concepts, our algorithm increases the average Spearman correlation from 0.283 to 0.389. Our debiasing algorithm also improves the generalization in prediction of the image labels. The debiasing also improves the top-5 accuracy of predicting the labels from 39.5% to 49.3%.

To show that our proposed debiasing accurately ranks the concepts in terms of their explanation of the predictions, we use the RemOve And Retrain (ROAR) framework (Hooker et al., 2019). In the ROAR framework, we sort the concept using the scores $E[\mathbf{d}|\boldsymbol{x}_i] - \frac{1}{n} \sum_{i=1}^{n} E[\mathbf{d}|\boldsymbol{x}_i]$ in the ascending order. Then, we mask (set to zero) the least explanatory $x\%$ of the concepts using the scores and retrain the $\boldsymbol{g_\theta}(\boldsymbol{d})$ function. We perform the procedure for $x \in \{0, 10, 20, \ldots, 80, 90, 95\}$ and record

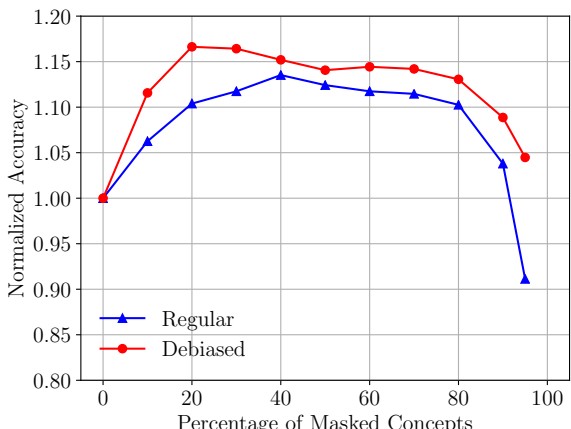

Figure 4: The ROAR evaluation: We mask $x\%$ of the concepts that are identified by the methods as less explanatory of the labels and retrain the $g_{\boldsymbol{\theta}}(\cdot)$ function. We measure the change in the accuracy of predicting the labels $\mathbf{y}$ as we increase the masking percentage. For better comparison of the trends, we have normalized them by their first data point ($x = 0\%$).

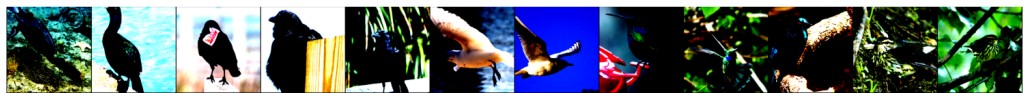

Figure 5: Twelve example images where the debiasing helps. A common pattern is that, the image context has either prevented or misled the annotator from accurate annotation of the concepts. From the left to right, the birds are 'Brandt Cormorant', 'Pelagic Cormorant', 'Fish Crow', 'Fish Crow', 'Fish Crow', 'Ivory Gull', 'Ivory Gull', 'Green Violetear', 'Green Violetear', 'Cape Glossy Starling', 'Northern Waterthrush', 'Northern Waterthrush'.

the testing top-5 accuracy of predicting the labels $\boldsymbol{y}$. We repeat the ROAR experiments for 3 times and report the average accuracy as we vary the masking percentage.

Figure 4 shows the ROAR evaluation of the regular and debiased algorithms. Because the debiased algorithm is more accurate, for easier comparison, we normalize both curves by dividing them by their accuracy at masking percentage $x = 0\%$. An immediate observation is that the plot for debiased algorithm stays above the regular one, which is a clear indication of its superior performance in identifying the least explanatory concepts. The results show several additional interesting insights too. First, the prediction of the bird species largely rely on a sparse set of concepts as we can mask $95\%$ of the concepts and still have a decent accuracy. Second, masking a small percentage of irrelevant concepts reduces the noise in the features and improves the generalization performance of both algorithms. Our debiased algorithm is more successful by being faster at finding the noisy features before $x = 20\%$ masking. Finally, the debiased accuracy curve is less steep after $x = 80\%$, which again indicates its success in finding the most explanatory concepts.

**Qualitative analysis of the results.**  In Figure 5, we show 12 images for which the $\widehat{\mathbf{d}}$ and $\mathbf{c}$ are significantly different. A common pattern among the examples is that the context of the image does not allow accurate annotations by the annotators. In images 3, 4, 5, 6, 7, 11, and 12 in Figure 5, the ten color-related concepts listed at the beginning are all set to 0.5, indicating that the annotators have failed in annotation. However, our algorithm correctly identifies that for example Ivory Gulls do not have green-colored backs by predicting $\widehat{c} = 0.08$ which is closer to $\widehat{c}(\mathbf{y}) = 0.06$ than the true $c = 0.5$.

Another pattern is the impact of the color of the environment on the accuracy of the annotations. For example, the second image from the left is an image of Pelagic cormorant, whose back and upper tail colors are unlikely to be green with per-class average of $0.12$ and $0.07$, respectively. However, because of the color of the image and the reflections, the annotator has assigned $1.0$ to both of 'has back color::green' and 'has upper tail color::green' concepts. Our algorithm predicts $0.11$ and $0.16$ for these two features respectively, which are closer to the per-class average.

Table 2: Examples of differences between regular and debiased algorithms in ranking the concepts.

| Image | Top 15 Concepts |
|---|---|
| 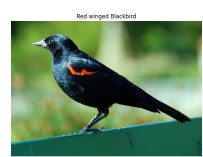 | **Debiased**: has throat color::black, has head pattern::plain, has forehead color::black, has breast color::black, has underparts color::black, has nape color::black, has crown color::black, has primary color::black, has bill color::black, has belly color::black, has wing color::orange, has breast pattern::solid, has upperparts color::orange, has wing pattern::multi-colored, has bill length::about the same as head
**Regular**: has primary color::black, has wing color::black, has throat color::black, has upperparts color::black, has breast color::black, has primary color::blue, has underparts color::black, has belly color::black, has back color::black, has nape color::black, has upperparts color::blue, has tail pattern::solid, has crown color::blue, has under tail color::black, has forehead color::blue |
| 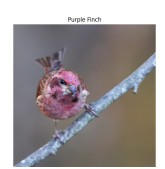 | **Debiased**: has primary color::red, has crown color::red, has forehead color::red, has throat color::red, has nape color::red, has breast color::red, has underparts color::red, has belly color::red, has forehead color::rufous, has upperparts color::red, has crown color::rufous, has nape color::rufous, has wing pattern::multi-colored, has primary color::rufous, has throat color::rufous
**Regular**: has underparts color::grey, has breast color::grey, has belly color::grey, has belly pattern::multi-colored, has breast pattern::multi-colored, has nape color::grey, has bill length::shorter than head, has breast color::red, has throat color::grey, has upperparts color::red, has back pattern::multi-colored, has underparts color::red, has primary color::grey, has belly color::red, has throat color::red |
| 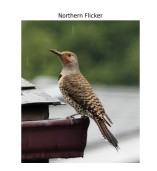 | **Debiased**: has bill length::about the same as head, has belly pattern::spotted, has underparts color::black, has bill shape::dagger, has breast color::black, has belly color::black, has breast pattern::spotted, has back pattern::spotted, has wing pattern::spotted, has nape color::red, has back color::black, has tail pattern::spotted, has under tail color::black, has upper tail color::black, has primary color::buff
**Regular**: has primary color::brown, has wing color::brown, has upperparts color::brown, has crown color::brown, has back color::brown, has forehead color::brown, has nape color::brown, has throat color::white, has breast pattern::spotted, has under tail color::brown, has breast color::white, has belly color::white, has underparts color::white, has upper tail color::brown, has breast color::brown |
| 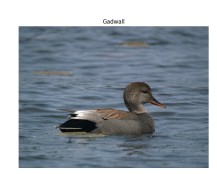 | **Debiased**: has shape::duck-like, has bill shape::spatulate, has size::medium (9 - 16 in), has bill length::about the same as head, has throat color::buff, has underparts color::brown, has breast color::brown, has belly pattern::spotted, has crown color::brown, has primary color::brown, has belly color::brown, has nape color::brown, has forehead color::brown, has upperparts color::brown, has belly color::purple
**Regular**: has belly color::grey, has underparts color::grey, has breast color::grey, has belly color::pink, has belly color::rufous, has underparts color::purple, has belly color::purple, has underparts color::pink, has throat color::grey, has primary color::grey, has belly color::green, has underparts color::rufous, has underparts color::green, has belly color::iridescent, has forehead color::grey |
| 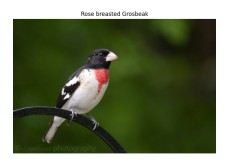 | **Debiased**: has throat color::black, has forehead color::black, has crown color::black, has primary color::black, has nape color::black, has breast color::red, has belly color::white, has underparts color::white, has underparts color::red, has back color::black, has primary color::white, has bill shape::cone, has breast pattern::multi-colored, has primary color::red, has upperparts color::black
**Regular**: has nape color::black, has primary color::black, has nape color::rufous, has primary color::red, has wing color::orange, has nape color::red, has breast color::red, has crown color::black, has upperparts color::orange, has crown color::red, has underparts color::white, has upperparts color::rufous, has forehead color::black, has throat color::red, has back color::purple |
| 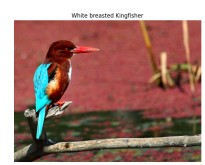 | **Debiased**: has wing color::blue, has upperparts color::blue, has crown color::rufous, has primary color::blue, has bill color::rufous, has back color::blue, has under tail color::blue, has bill color::red, has upper tail color::blue, has wing pattern::multi-colored, has nape color::rufous, has crown color::brown, has belly color::brown, has nape color::brown, has underparts color::brown
**Regular**: has primary color::red, has throat color::red, has underparts color::red, has breast color::red, has forehead color::red, has crown color::rufous, has crown color::red, has nape color::red, has belly color::red, has primary color::rufous, has throat color::rufous, has forehead color::rufous, has wing color::red, has nape color::rufous, has underparts color::rufous |

In Table 2, we list six examples to show the superior accuracy of the debiased CBM in ranking the concepts in terms of their explanation power. Moreover, in Table 3 in the appendix, we list the top and bottom concepts in terms of their predictive uncertainty in our debiased CBM.

## 5    CONCLUSIONS AND FUTURE WORKS

Studying the concept-based explanation techniques, we provided evidences for potential existence of spurious association between the features and concepts due to unobserved latent variables or noise. We proposed a new causal prior graph that models the impact of the noise and latent confounding fron the estimated concepts. We showed that using the labels as instruments, we can remove the impact of the context from the explanations. Our experiments showed that our debiasing technique not only improves the quality of the explanations, but also improve the accuracy of predicting labels through the concepts. As future work, we will investigate other two-stage-regression techniques to find the most accurate debiasing method.

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

## A  HIGH-NOISE SYNTHETIC EXPERIMENTS

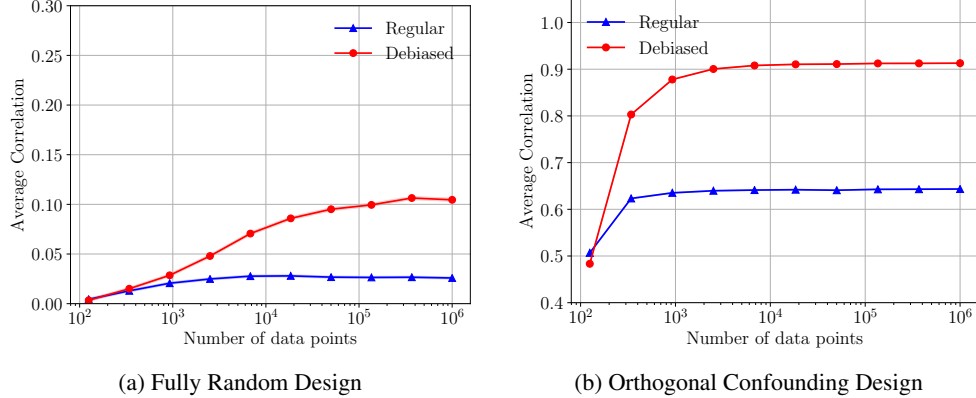

(a) Fully Random Design    (b) Orthogonal Confounding Design

Figure 6: Correlation between the estimated concept vectors and the true discriminative concept vectors as the number of data points grow. Notice the different ranges of the y-axes and the logarithmic scale of the x-axes.

| Certainty Level | Concepts |
| --- | --- |
| Most Uncertain | (194) has nape color::black, (260) has primary color::black, (164) has forehead color::black, (7) has bill shape::all-purpose, (132) has throat color::black, (305) has crown color::black, (133) has throat color::white, (150) has bill length::about the same as head, (8) has bill shape::cone, (152) has bill length::shorter than head |
| Least Uncertain | (216) has wing shape::tapered-wings, (215) has wing shape::broad-wings, (217) has wing shape::long-wings, (214) has wing shape::pointed-wings, (77) has tail shape::fan-shaped tail, (213) has wing shape::rounded-wings, (79) has tail shape::squared tail, (83) has upper tail color::purple, (82) has upper tail color::iridescent, (75) has tail shape::rounded tail |

Table 3: The top 10 most and least uncertain concepts identified by the debiased algorithm.

