# OpenReview forum: "Debiasing Concept-based Explanations with Causal Analysis"
_ICLR.cc/2021/Conference — ICLR 2021 Poster_

### Official Review · AnonReviewer1 · 2020-10-28
**Interesting paper whose exposition needs improvement**

**Rating:** 6
**Confidence:** 4

**Review:**

Overview of the Paper
This paper proposes a procedure to 'debias' and account for confounding while using 'concepts' as interpretations for black-box models. The approach proposes a causal graph, and then notes based on the proposed graph that sample labels satisfy conditions required of an instrumental variable. Given an instrumental variable, then they propose a multistage approach to debias the interpretations. In general, this paper provides a useful approach for helping to generate concept explanations that are not confounded with the label or other features.

General Feedback / Quality
This paper contains several interesting ideas. I believe this is the first paper to bring the idea of using instrumental variables as debiasing approaches for explanations. Having this said, I cannot recommend the paper in its current form because the exposition makes the paper a tough read. For example, this paper builds on the concept bottleneck approach of Koh et. al. from ICML 2020. However, no contained overview of that approach is given. The only reason I am able to follow the discussion is because I previously read that the Koh et. al. paper and I am generally familiar with instrumental variables. A more damning issue is that the paper does not explicitly set out the question/problem that it seeks to address. As it stands, figure 1 provides a vague motivation, but it is not clearly contextualized. The dataset and the discussion of concepts is not explicitly presented, so one is left to guess how the dataset was processed etc. I have worked with the CUB birds dataset before, so I am familiar with the concepts and data. My takeaway is that the paper requires a substantial rewrite that hand-holds a reader through the work. I document some of the other questions I had in the latter parts of this review.

Originality
The use of instrumental variable methods for explanation debiasing is new and original. The intersection of causal inference and post-hoc interpretability is a new and emerging area, so this work is one of the first to set out this path. There is other work in this area, but these do not address concept bottleneck models.

Significance
As noted above, the main idea and goal of this work is important. Overall, it is important to obtain explanations from models that are not confounded with other variables. It is also likely important for diagnosing bugs.

Clarity and Writing
The paper is mostly free of typos, but as it stands it is quite difficult to read. Several important concepts are not properly introduced, and a substantial rewrite is likely needed. See the overview for a discussion of some of these issues.

Some other questions/concerns

- Conceptual Confusion: Throughout this paper, it is unclear to how the methods here should be operationalized. Do the authors envision that instead of computing standard TCAVs or concepts, these concepts would first be debiased via the an instrumental variable following the protocol described here?

- The causal graph: I am not an expert on causal inference but I wanted to get the authors' opinion on whether the proposed causal graphs are standard in that literature. Take fig. 2a., the causal graph says that the labels cause the concepts, which then cause the features. However, could this also be the other way? Here is an example: is it the case that calling an object a chair causes the object to have four legs or does an object with four legs cause or induce a label chair? I think the second graph would not fit under the scheme presented, but I am curious if this is the case. Primarily, I don't think the model labels can be used as instruments in such setting, but perhaps the framework is still appropriate?

- The evaluation. I think the paper missed an opportunity the show a wow factor here. From the CUB dataset, it seems like the take away is that the debiased concept interpretations can identify a spurious correlation that a standard concept interpretation cannot? If this is the case, then that is a substantial positive for the approach. Instead of the ROAR evaluation, the authors should have created models with artificial spurious correlations and show that standard concept methods are unhelpful.

- The ROAR approach used in this work is not an appropriate form of evaluation. I agree with the authors that explanation evaluation is difficult but the ROAR metric does not do this. The method retrains a model after masking out some highly important features, and then interprets the retrain model in comparison to the original one. This is not a comparison, especially when the models that are being compared a not convex. There is no guarantee that a retrained model relies on the same set of rules as the original model.  I would suggest the authors create toy settings (with real or toy datasets) where the DGP can be easily manipulated and show the promise of their approach that way.

------
Post Reponse Update
-----
Thanks to the authors for their updates. I have updated my score by 1 point here. I believe the exposition in the paper could still be improved at this point. In general, this work provides an interesting use of IV techniques for interpreting black box models.

---

> ### Author Response · Authors · 2020-11-16
> **Good Questions**
>
> **RE: Clarity/writing**: We have overviewed the CBMs in the problem statement at the beginning of Section 2. We have provided a background on the instrumental variables in Section 2.2. We added a new paragraph after the problem statement to concretely state the problem that we want to solve in a formal language. Please let us know which aspects are not covered in these three segments.
>
> **RE: Conceptual Confusion**: Your understanding is correct.
>
> **RE: The causal graph**: Every rigorous causal analysis (in both Pearl and Rubin’s causal inference frameworks) relies on a set of "untestable" assumptions (encoded as a causal prior graph in Pearl's framework). Our work is not an exception to this rule. In the experiments, we show that our causal prior assumption does work.
>
> **RE: The evaluation**: As researchers in the causality domain, we know that the causal results are always rooted in the causal prior assumption and should not surprise anyone. Moreover, in the synthetic experiments, we have done exactly what you suggest us to do.
>
> **RE: The ROAR approach**: There is confusion here. Our CBM consists of two models $\mathbf{x} \to \mathbf{c}$ and $\mathbf{c} \to \mathbf{y}$. During our ROAR evaluation, we fix the $\mathbf{x} \to \mathbf{c}$ network and only retrain the $\mathbf{c} \to \mathbf{y}$ network. Given this and according to the reasoning in the ROAR paper, the evaluation is rigorous.

---

### Official Review · AnonReviewer3 · 2020-10-30
**Good paper, with a clever use of instrumental variables. I liked the results but I would have liked a clear description of the scope in which the method can be applied.**

**Rating:** 7
**Confidence:** 3

**Review:**

Abstract: The paper proposed a way to learn unbiased (debiased) concept-based explainable models in the presence of unobserved confounders by the use of labels as instrumental variables. The proposed algorithm has 3 main steps: (1) regresses concept labels from the final labels (2) replace the original concepts with the learnt concepts and learn a model of debiased concepts as a function of features (3) predict label as a function of debiased concepts. The authors show in the experimental section that their training method captures the most salient concepts using the ROAR (Remove and Retrain) evaluation framework much better than the vanilla (non de-biased) approach.

Pros:
- Clarity: The paper is quite well-written and the explanations are clear.
- Significance/Impact: The ability to explain modeling prediction is quite important and furthermore it can lead to better generalization and robustness
- Experimental design: I really liked the experimental section, very clear and well-done

Cons:
- The causal graph assumption
- Motivation: I would have spent a bit more time on explaining the motivation of the work and the scope that it adresses. For example, the causal graph assumption that makes labels to be the causal root of the graph, works well in image classification, but not in learning-to-control applications, such as recommender systems where the label is the reward and is the final consequence of the causal chain.

---

> ### Author Response · Authors · 2020-11-16
> **Agreed!**
>
> Thank you for your encouraging comments.
>
> 1. Similar to all rigorous works based on both Pearl and Rubin’s causal inference frameworks, we are bound to our prior assumptions. We have done our best to show that our debiasing algorithm works in more realistic settings by including the $\mathbf{y} \to \mathbf{x}$ edge.
> 2. We will expand the motivation and clarify the scope soon.

---

### Official Review · AnonReviewer5 · 2020-11-06
**Concerns about the model**

**Rating:** 5
**Confidence:** 4

**Review:**

The focus of the work is on model interpretability using concept-based explanation. The authors consider the issue of concepts being correlated with confounding information in the features. They propose a causal graph for representing the system and use instrumental variable methods to remove the impact of unobserved confounders. The proposed method is evaluated on synthetic and real data.

I have the following comments:

(1) The authors provide a graphical modelings for the setup of the problem. They assume that: 1. There is an "unconfounded concept" generated only from the label, 2. There is no confounding effect on the label.

Unfortunately, no arguments is provided for justifying the conditional independence assumptions in the model. Any edge that is removed from a graphical model implies conditional independency assumptions and they should be carefully justified, specially since the topic of the work is interpretability. This includes the conditional independencies above as well as the way it is assumed that variable c is generated.

(2) The model in this work is different from an IV model: One of the main requirements of the IV framework is exclusion restriction which requires that the effect of the instrument variable on the outcome should be only through the treatment variable. In the proposed model, variable y is also directly connected to variable x. Also, d is not confounded or observed, which again make the model different from IV model. Therefore, the model in this work does not represent the IV model, and y is not a valid IV, although it seems that that was actually not used in the approach. Only an independence assumption is actually used in the approach, which as mentioned above, is not justified.

(3) It is not quite clear what exactly the variable d represents compared to variable c, and what is its interpretation.

(4) In the synthetic simulations, the value of the variance chosen, specially for noises is very small. It is important to see the performance for larger values for the variance of the noises.

---

> ### Author Response · Authors · 2020-11-16
> **Clarifications + A New Synthetic Experiment**
>
> 1. Three points:
>     1. First, we need to clarify that our debiasing method removes the noise in addition to the confounding effects.
>     2. Second, Figure 1 provides a strong piece of evidence for the existence of confounding and noise effects described by our causal prior graph.  In CBMs we have three levels of variables in the increasing order of details: (1) Labels, (2) Concepts, (3) Images. While the labels are only about the identity of the object in the picture, concepts and Images contain extra information about the context. The phenomenon shown in Figure 1 alludes to existence of confounding and noise that is independent of the labels.
>     3. Third, in both Pearl and Rubin’s causal frameworks, we always start with expert knowledge as prior knowledge (untestable assumption) and say that our results are valid when the assumptions hold.
> 2. Three points:
>     1. Based on theory of causal inference, given our causal prior graph, we prove the correctness of our debiasing algorithm both graphically and mathematically, regardless of the connections to the instrumental variables.
>     2. As you pointed out, our setting is not a classical instrumental variable setting. Our use of the term “Instrumental Variable” is appropriate because we show that the commonly known multi-step IV "_technique_" can be used in our causal prior graph. That is the key point of this paper. We will clarify the intricate distinction in the paper.
>     3. If you strongly think that our use of the term is misleading, we can drop the “Instrumental Variable” phrase from the title and only describe the connections to the multi-step IV estimation technique. This won’t have any impact on the methodological contributions and the experimental results of the paper.
> 3. The variable $\mathbf{d}$ denotes the "discriminative concepts." We provide the mathematical definition of $\mathbf{d}$ in our structural equations on page 3. According to our definition $\mathbf{d}$  contains all the information in $\mathbf{c}$  that are relevant to $\mathbf{y}$  and excludes the confounding information in the context $\mathbf{u}$ .
> 4. Upon your suggestion, we added Figure 6 in the appendix with 5 times larger noise standard deviation ($\sigma=0.1$). The results confirm our claims. Thank you for the suggestion.

---

> ### Author Response · Authors · 2020-11-18
> **Update**
>
> After deliberation between ourselves, we decided to prevent confusion between the IV _technique_ and _setting_.  In the new revision, in response to your points, we use the phrase "_two-stage regression technique from instrumental variables literature_" to avoid confusion. We have updated the title, abstract, introduction, conclusion, and part of the methodology section that refers to IVs.

---

### Official Review · AnonReviewer2 · 2020-11-08
**Interesting direction but not convincing enough**

**Rating:** 4
**Confidence:** 3

**Review:**

Summary: The paper studies the causal nature of concept explanations. Specifically, the authors treat labels as instrumental variables to then debias explanations and improve predictive performance as well.

Strengths
- Figure 2 was helpful for understanding the contributions of this work. However, it is not clear if \hat_d captures the same concepts that c alone would. If you could motivate using \hat_d with a pictorial example of where c, the concepts alone, fail that would be helpful.
- The use of ROAR for concepts is clever and does show the utility of the method, but additional experimentation to show the concepts captured align with human intuition would have been nice. At the minimum, showing how the concepts recovered by the proposed method and CBM differ would be helpful.

Weaknesses
- While both experiments (synthetic and BIRDs) show the method's utility, it would have been nice to see experiments on other datasets. OAI perhaps like in Koh et al.
- The connection to Yeh et al. is not clear to me. What is the notion of completeness in the proposed method?

Question
- Can you extract uncertainty estimates for concepts from Equation 5?
- Can you please explicitly the utility of your method in the linear Gaussian case? It seems as though using \hat_d for the concepts simply recovers the independent concept case from Koh et al.

---

> ### Author Response · Authors · 2020-11-16
> **Clarifications + 2 New Results**
>
> We appreciate your constructive comments.
> * The variable $\mathbf{d}$ denotes the "_discriminative concepts_." We provide the mathematical definition of $\mathbf{d}$ in our structural equations on page 3. According to our definition $\mathbf{d}$ contains all the information in $\mathbf{c}$ that are relevant to $\mathbf{y}$ and excludes the confounding information in the context $\mathbf{u}$.
> * We added Table 2 upon your suggestion for deep dive into the results. Thank you. We hope our response is satisfactory.
> * We did not go fishing for datasets. Our algorithm worked really well on the first dataset (CUB) that we tried. Instead of working on another dataset and increase the sample size of datasets from "1" to "2", we preferred to provide insights about our algorithm by diving deeper into the results.
> * The connection to Yeh et al is not central to this work. In our causal prior graph, the $\mathbf{y} \to \mathbf{x}$ edge captures the concept completeness. Yeh et al is the paper that studies the "concept completeness" phenomenon and it is our scientific duty to cite and discuss it.
> * We have listed the concepts in terms of the increasing uncertainty in Table 3 in Appendix A. Thank you again for this suggestion.
>
> Overall, upon your suggestions, we have added two new sets of results. We hope that the changes help the results to be more convincing.

---

### Decision · Program_Chairs · 2021-01-07
**Final Decision**

**Decision:**

Accept (Poster)

**Comment:**

The paper has merits on providing a particular way of understanding a prediction model based on auxiliary data (concepts). I have a generally more positive view of it, aligned with the higher-scoring reviews. However, I feel a bit uncomfortable of framing it as "causal" in the sense it does not aim to provide any causal predictions, but it is more of a smoothing method for capturing signal contaminated with "uninteresting" latent sources - this is more akin to regression with measurement error (see e.g. Carroll, Ruppert and Stefanski's "Nonlinear regression with measurement error") where, like in this paper, different definitions of "instrumental variables" also exist and are different from the causal inference definition. I can see though why we may want to provide a causal interpretation in order to justify particular assumptions, not unlike interesting lines of work from Scholkopf's take on causality. The paper can be strengthened by some further discussion on the assumptions made about additivity on equations (2) and (3), which feel strong and not particularly welcome in many applications.

The proposed title is still a bit clunky, I feel that the two-stage approach is less important than the structural assumptions made, perhaps a title emphasizing the latter rather than the former would be more promising.